



# Effect of individual blade pitch angle misalignment on the remaining useful life of wind turbines

Matthias Saathoff[1], Malo Rosemeier[2], Thorsten Kleinselbeck[3], and Bente Rathmann[1]

[1]P. E. Concepts GmbH, Wiener Str. 5, 28359 Bremen, Germany
[2]Department of Rotor Blades, Fraunhofer IWES, Fraunhofer Institute for Wind Energy Systems, Am Seedeich 45, 27572 Bremerhaven, Germany
[3]WIND-consult GmbH, Reuterstr. 9, 18211 Bargeshagen, Germany

**Correspondence:** Matthias Saathoff (matthias.saathoff@p-e-c.com)

**Abstract.** An empirical data set of laser-optical pitch angle misalignment measurements on wind turbines was analyzed, and showed that 38% of the turbines have been operating outside the accepted aerodynamic imbalance range. This imbalance results from deviations between the working pitch angle and the design angle set point. Several studies have focused on the consequences of this imbalance for the annual energy production (AEP) loss and mention a possible decrease in fatigue budget, i.e., remaining useful life (RUL). This research, however, quantifies the effect of the individual blade pitch angle misalignment and the resulting aerodynamic imbalance on the RUL of a wind turbine. To this end, several imbalance scenarios were derived from the empirical data representing various individual pitch misalignment configurations of the three blades. As the use case, a commercial 1.5 MW turbine was investigated which provided a good representation of the sites and the turbine types in the empirical data set. Aeroelastic load simulations were conducted to determine the RUL of the turbine components. It was found that the RUL decreased in most scenarios, while the non-rotating wind turbine components were affected most by an aerodynamic imbalance.

## 1 Introduction

During the manufacture of wind turbine blades, pitch bearings, and hubs, a reference mark is positioned at the bolt circle diameter, which is used to position the blade at the hub with respect to the rotor plane. Manufacturing tolerances of the bolt positions at the blade root, the pitch bearing, and the hub flange (Elosegui et al., 2018), human errors while marking the reference at the blade root, pitch bearing, or hub (Elosegui and Ulazia, 2017), as well as the correct positioning of the blade with respect to the hub (Cacciola et al., 2016) seem to be the main root causes for an aerodynamic imbalance, also referred to as aerodynamic asymmetry, of rotors observed in the field, cf. Grunwald et al. (2015). The imbalance results from the blades' misalignments between the actual working pitch angle and the design pitch angle set point. During the calculation of the design





loads, the aerodynamic imbalance must be taken into account according to the design standards and guidelines, cf. DIBt (1993); DS (1992); GL (2010).

Aerodynamic imbalance causes not only a loss in energy yield, but also an increase in vibration and rotor speed fluctuations, as well as loads on most turbine components and, in turn, a shorter turbine life (Hyers et al., 2006; Elosegui et al., 2018; Astolfi, 2019). Kusnick et al. (2015) have shown that the load on the main shaft increases significantly as a consequence of a pitch misalignment implemented in one blade. A turbine in the field is subjected to increased loads until the pitch misalignment is detected and corrected.

Wind farm operators are motivated to reduce the levelized cost of energy (LCoE) of their wind turbines. First, the loss in energy yield due to an aerodynamic imbalance increases LCoE. Second, on most sites, the fatigue budget is not fully utilized during the design lifetime of a turbine. The LCoE can be decreased when the energy yield over the lifetime is increased. To achieve this, the lifetime of the turbine can be extended until the fatigue budget is exhausted. Therefore, the quantitative effect of the aerodynamic imbalance on the remaining useful life (RUL) is of economic importance for the operator.

The pitch angle misalignment can be measured by photometric means, where a camera is placed directly below a blade pointing downwards to the ground, i.e., six o'clock position (Grunwald et al., 2015). Another technique uses laser distance measurements of the pressure side surface of the blade to determine the relative pitch misalignment between the blades while the turbine is in power production mode (windcomp GmbH, 2020). Wang et al. (2009) applied a laser tracker to reconstruct the 3D shape of the rotor blade. Elosegui and Ulazia (2017) used a laser tracker to detect the aerodynamic imbalance. Elosegui et al. (2018) pointed out that such a technique fails to detect the absolute pitch misaligment, which was key to ensuring energy yield improvement and avoiding changes in turbine lifetime.

Kusiak and Verma (2010) proposed a data-driven method to detect the aerodynamic imbalance. Niebsch and Ramlau (2014) proposed an algorithm to detect the rotor imbalance. Cacciola et al. (2016) presented a method based on a neural network to detect the aerodynamic imbalance. Bertelè et al. (2018) presented an algorithm to correct the aerodynamic imbalance.

Rosemeier and Saathoff (2020b) found that a manufacture-induced blade shape distortion, e.g., blade twist, reduced the energy yield and increased the RUL.

To the authors' knowledge, the effect of the aerodynamic imbalance on the RUL has been mentioned in literature, but not been quantified. To this end, this research analyzes empirical data of pitch angle misalignments encountered in the field. This analysis is then used to derive representative aerodynamic imbalance scenarios. As a use case, a commercial turbine sited in Northern Germany is chosen for the assessment of the RUL of the turbine.

This article is structured as follows: Chap. 2 describes how the measurement data was acquired and presents the empirical data. Furthermore, the imbalance scenarios are derived and the simulation model is described. Chap. 3 presents the results of the RUL calculation. Chap. 4 discusses the effect of pitch angle misalignment and the representativity of this research. Finally, a conclusion summarizes the findings of this research.



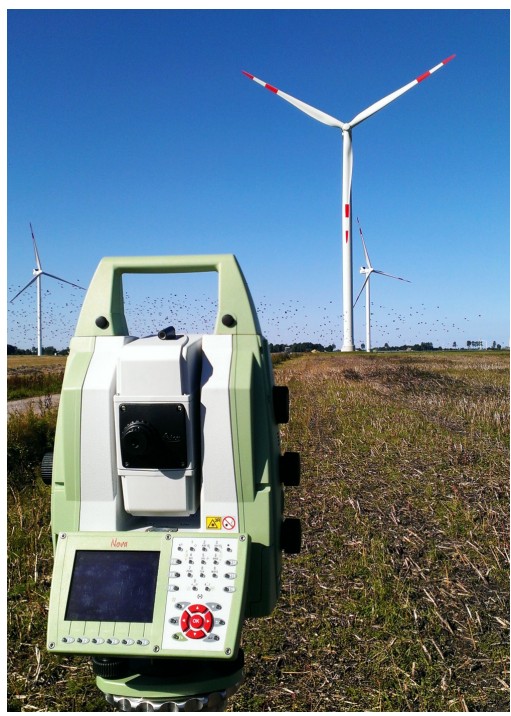

**Figure 1.** Measurement setup: tachymeter and turbine at a standstill.

## 2 Methods

### 2.1 Measurement of blade pitch angle misalignment

The blade pitch angle measurements were carried out based on a laser-optical method where the three-dimensional shape of the blade was captured with reference to a coordinate system at the center of the hub (Kleinselbeck and Hagedorn, 2015). A tachymeter (Fig. 1) detected a cloud of points representing the outer pressure side shape of the rotor blade (Fig. 2). For the measurement, the pitch axis of the blade to be captured is in the six o'clock and the fine pitch positions. The measured 3D coordinates were transformed into surfaces and subsequently compared with a target airfoil shape using an in-house software

(Fig. 3). To this end, the cross sectional shapes of the three blades were reproduced at maximum chord length, while the pitch axis was used as a reference to project the shapes onto a common coordinate system. From these shapes, the angle between the three blades within the rotor could be determined with an accuracy of $\pm 0.1°$. When the target airfoil shape, its local twist angle, and the target pitch angle of the turbine's blade type were known, it was possible to derive the absolute individual pitch angle misalignment. If this was not the case, the average of the measured pitch angle misalignments across the three blades

was used as the target pitch angle, or an individual decision for the target pitch angle was made.

Empirical data on blade pitch angle measurements obtained between 2013 and 2021 in Europe, North and South America were analyzed. The data set of more than 1,100 turbines was filtered to 195 turbines representing a fleet of turbine types that

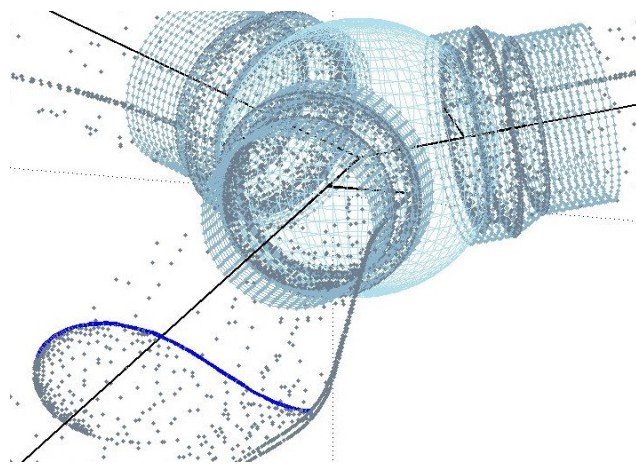

**Figure 2.** Identification of chord shape from measured coordinates.

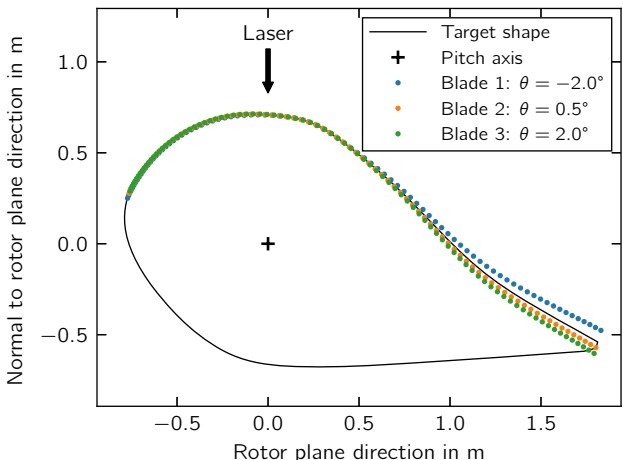

**Figure 3.** Example of measured shapes of the three blades within the rotor; view from the blade tip.

need to be assessed for a lifetime extension today or within the next few years and are located mainly in Northern Europe. The set contained 22 combinations of turbine and blade types in the power range between 0.9 MW and 2.5 MW.

Fig. 4 shows the minimum and maximum measured individual pitch misalignment $\theta$ of the rotor for each of the measured turbines within the filtered data set. At first, the measurement data was grouped according to the aerodynamic imbalance $\Delta\theta = \theta_{max} - \theta_{min}$, and furthermore, sorted according to $\theta_{min}$. The individual misalignment of each blade between $\theta_{min}$ and $\theta_{max}$ was not further taken into account in this research.



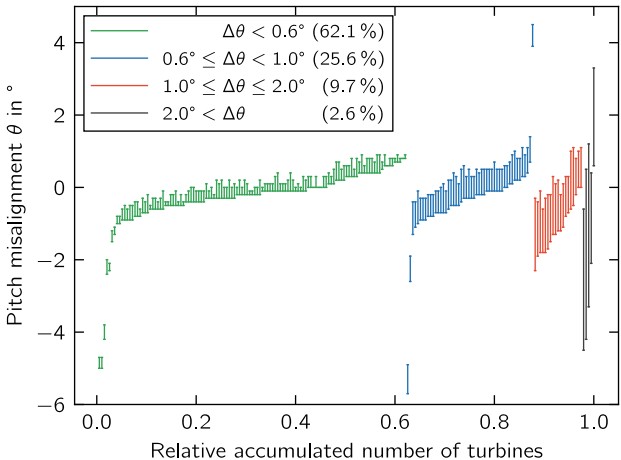

**Figure 4.** Measured minimum and maximum pitch angle misalignment across analyzed wind turbines. A positive pitch misalignment $\theta > 0°$ refers to a misalignment toward feather with respect to the target pitch angle.

## 2.2 Modeling of aerodynamic imbalance

To assess the impact of aerodynamic imbalance on the RUL, the design imbalance was compared to several imbalance scenarios, which were derived from the pitch angle measurement data set.

  For the design situation, the aerodynamic imbalance is derived from relevant design standards and guidelines. While more recent standards such as IEC 61400-1 Ed. 2 to Ed. 4 (IEC, 1999, 2010, 2019) do not give any guidance as to the specifications regarding the rotor imbalance, older standards and guidelines specify explicit values. According to the guidelines from Ger-

manischer Lloyd (GL, 1989, 1993, 1999, 2004, 2010), the guideline from Deutsches Institut für Bautechnik (DIBt, 1993), and the DS 472 (DS, 1992), a pitch angle misalignment of $\theta = \pm 0.3°$ should be considered during design. As these standards and guidelines were commonly applied in the design of wind turbines, an aerodynamic imbalance of $\Delta\theta = 0.6°$ is a reasonable choice.

  Fig. 5 shows the measured data sorted according to the aerodynamic imbalance $\Delta\theta$ while using the same grouping as

in Fig. 4. The green group represents 62.1% of the turbines whose aerodynamic imbalance is within the limit of $\Delta\theta < 0.6°$ accepted by the design standards and guidelines. The next groups (blue and red) comprise 34.8% of the measured wind turbines with $0.6° \leq \Delta\theta \leq 2.0°$. Our research focused on these two groups. The subdivision into two groups was made to quantify the effect of different imbalance intensities. Imbalances of $\Delta\theta > 2.0°$ were considered to be outliers.

  Seven imbalance scenarios were defined to represent the blue and red group. The combinations of blade angle misalignments

in each scenario are summarized in Table 1. Scenario D represents the green group and the design situation. Scenario S1.1 and S1.2 represent the blue group and the scenarios S2.1 to S2.5 represent the green group. The combinations of $\theta_{min}$ and $\theta_{max}$ across the three blades were chosen to obtain the largest deviation in aerodynamic imbalance $\Delta\theta$ between the design scenario D and the respective imbalance scenario. Only two scenarios were selected to represent the blue group with $\Delta\theta = 1.0°$, i.e.,

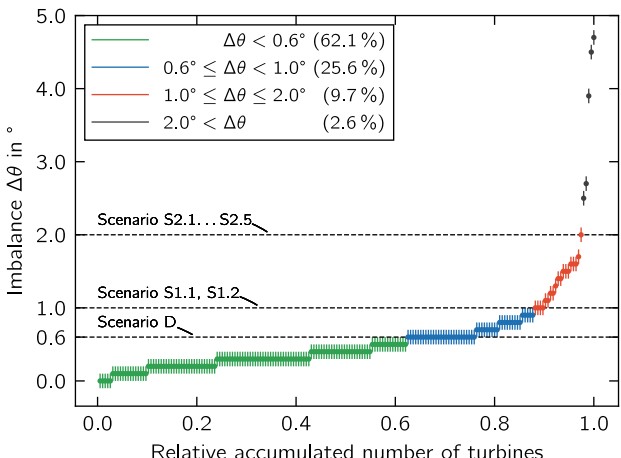

**Figure 5.** Aerodynamic imbalance derived from the measurement data set.

**Table 1.** Imbalance scenarios simulated.

| | Pitch angle misalignment $\theta$ in ° | | | |
| Scenario | Blade 1 | Blade 2 | Blade 3 | $\Delta\theta$ in ° |
| --- | --- | --- | --- | --- |
| D | -0.3 | 0.3 | 0.0 | 0.6 |
| S1.1 | 0.0 | -1.0 | -1.0 | 1.0 |
| S1.2 | 0.0 | 0.0 | -1.0 | |
| S2.1 | 0.0 | -2.0 | -2.0 | |
| S2.2 | 0.0 | 0.0 | -2.0 | |
| S2.3 | 1.0 | -1.0 | 0.0 | 2.0 |
| S2.4 | 0.0 | 0.0 | 2.0 | |
| S2.5 | 2.0 | 0.0 | 2.0 | |

$\theta < 0°$ refers to toward stall; $\theta > 0°$ refers to toward feather

S1.1 and S1.2, as they allow a direct comparison to the most severe scenarios to represent the red group, i.e., S2.1 and S2.2,
respectively.

### 2.3 Use case turbine and site

A wind turbine located at Bremervörde-Iselersheim, Germany, was modeled as an example. The wind turbine of the type
Südwind S70 is a variable-speed, pitch-controlled wind turbine with a rated power of 1.5 MW. The chosen wind site and
turbine type are a good representation of the average of the sites and turbine types analyzed in the empirical data set of pitch
angle misalignment measurements, see above. This turbine type was certified for a large number of blade types, rotor diameters





**Table 2.** Design parameters of the Südwind S70 turbine type.

| Parameter | Symbol | Value | Unit |
|---|---|---|---|
| Wind turbine class | - | III | - |
| Turbulence category | - | A | - |
| Rated power | $P_\mathrm{r}$ | 1.5 | MW |
| Rotor radius | $R$ | 35 | m |
| Hub height[1] | $h$ | 65 | m |
| Cut-in wind speed | $v_\mathrm{in}$ | 3.5 | $\mathrm{m\,s^{-1}}$ |
| Cut-out wind speed | $v_\mathrm{out}$ | 25.0 | $\mathrm{m\,s^{-1}}$ |
| Cut-in rotor speed | $\Omega_\mathrm{in}$ | 10.5 | $\mathrm{min^{-1}}$ |
| Rated rotor speed | $\Omega_\mathrm{r}$ | 19.0 | $\mathrm{min^{-1}}$ |
| Design lifetime | $T_\mathrm{d}$ | 20 | years |

[1] Marktstammdatenregister (2021)

and hub heights. The design loads were assumed for the turbine class IIIA according to IEC 61400-1 Ed. 2 (IEC, 1999). The average wind speed for the design situation is $v_\mathrm{ave} = 7.5\,\mathrm{m\,s^{-1}}$ and the reference turbulence intensity $I_\mathrm{ref} = 0.16$. This is a conservative estimate since some turbine components of the type S70 were designed to withstand the loads corresponding to wind turbine class II. The chosen design parameters of the turbine are given in Table 2.

The wind conditions on site were estimated according to Eurocode 1 (CEN, 2009) for the respective wind zone. The derived wind distribution was verified by a site assessment of a nearby wind farm. The effective turbulence on a turbine in the farm was approximated with turbulence class B according to IEC 61400-1 Ed. 3 (IEC, 2010). In this case, the effective turbulence is given by a reference turbulence intensity $I_\mathrm{ref} = 0.14$ using the Normal Turbulence Model. The wind speed at the site was characterized by an average wind speed of $v_\mathrm{ave} = 6.7\,\mathrm{m\,s^{-1}}$ using a Rayleigh distribution.

## 110   2.4   Aeroelastic load simulation

The use case turbine was modeled in the wind turbine simulation software openFAST v2.3.0 (NREL, 2020). AeroDyn (Moriarty and Hansen, 2005) calculated the aerodynamic loads taking the aeroelastic coupling with modal bodies into account. The turbulent wind fields were generated using TurbSim (Jonkman, 2009).

The tower structure was modeled taking into account parameters from the type certificate and related documents of the S70

turbine (ABH, 1999). The structural and aerodynamic properties of the SSP34 rotor blade type were considered in the model. A generic proportional-integral (PI) controller was applied for the pitch and generator torque controls. The maximum pitch speed of the S70 turbine was chosen to be $5\,^\circ\mathrm{s^{-1}}$. The electrical pitch drives were modeled with a second order lag element with a corner frequency of $1.2\,\mathrm{Hz}$ and a damping ratio of $\delta = 0.8$.





The load simulations were conducted for the design and the site wind conditions. In both simulation sets, the fatigue loads
were generated from Design Load Case (DLC) 1.2 and DLC 6.4 according to IEC 61400-1 (IEC, 2010). For both the design
and site load sets, the same random seeds were used for the wind fields to provide comparability. The wind speed bins were
each considered with a resolution of $1\,\mathrm{m\,s^{-1}}$ and $2.5\,\mathrm{h}$ simulation time. The availability of the turbine was assumed to be 100%.

## 2.5  Calculation of remaining useful life

The time series from the load simulation were post-processed with a rainflow counting algorithm (Madsen et al., 1990). From
the load spectra, damage-equivalent loads (DELs) were calculated according to Hayman (2012). Assuming a linear relationship
between mechanical stresses and external loads on the one hand and neglecting mean stress sensitivity (Saathoff and Rosemeier,
2020) on the other, the fatigue damage $D_{\mathrm{rel}}$ can be expressed as

$$D_{\mathrm{rel}} = \left( \frac{M_{\mathrm{eq}}^{\mathrm{a,s}}}{M_{\mathrm{eq}}^{\mathrm{a,d}}} \right)^{m}, \tag{1}$$

where $M_{\mathrm{eq}}^{\mathrm{a,s}}$ denotes the DEL of the site condition, $M_{\mathrm{eq}}^{\mathrm{a,d}}$ denotes the DEL of the design condition, and $m$ denotes the negative
inverse S-N curve exponent of the respective component's material. If we further assume that there is only a single relevant
load situation and that each component was designed for exactly the design lifetime $T_{\mathrm{d}}$, the RUL is calculated as

$$T_{\mathrm{RUL}} = T_{\mathrm{d}} \left( \frac{1}{D_{\mathrm{rel}}} - 1 \right), \tag{2}$$

cf. Rosemeier and Saathoff (2020a).

The remaining useful life is determined for the turbine components that are critical to the structural integrity of the wind
turbine, i.e., the blade root, blade bolts, hub, rotor shaft, main frame, and tower base. In general, the component with the
shortest $T_{\mathrm{RUL}}$ limits the possible lifetime extension of the turbine.

## 3  Results

### 3.1  Remaining useful life

First, we consider the RUL in the design scenario D (Fig. 6) representing the green group of the measurement data set. The
blade bolts limit the RUL of the use case turbine giving a $T_{\mathrm{RUL}} = 3\,\mathrm{years}$. Since the blade bolts could in principle be exchanged,
the critical component is the hub, which limits the RUL to $T_{\mathrm{RUL}} = 4.2\,\mathrm{years}$.

Second, we consider the effect of the aerodynamic imbalance scenarios, which represent the blue and red group, on the
RUL. Here, we assume that the aerodynamic imbalance is not corrected during the entire lifetime of the turbine. Scenario
S2.1 shows the largest decrease in RUL. Three component groups show similar RULs when compared to each other. The first
group contains components in the vicinity of the blade connection, i.e., the blade root, the blade bolts and the hub. Here, the
smallest RUL ranges between 2.9 years and 5.9 years. The second group contains the rotor shaft, which also includes the bolted

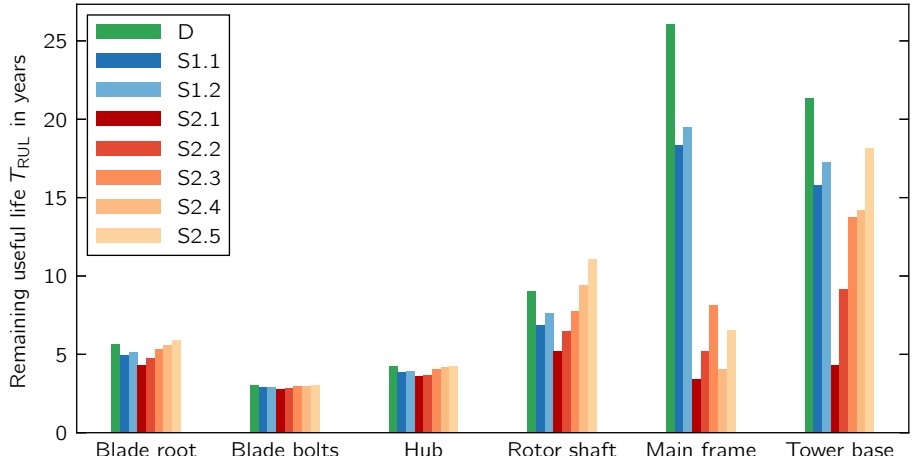

**Figure 6.** Remaining useful life (RUL) of components in design (D) situation and imbalance scenarios (S).

connection to the hub. For these components, the RUL ranges from 6.2 years to 11.1 years. The main frame and tower base represent the third group with RUL values between 3.4 years and 26.1 years.

The third group is affected the most in the scenarios representing the red group with $\Delta\theta = 2°$. In scenario S2.1, the main frame, in particular, limits the RUL, whereas the hub would be limiting in the design scenario D. Since the fatigue damage is linearly proportional to the lifetime, the loss in RUL can be related to one operational year by dividing $T_{RUL}$ by the design lifetime. Thus, for each operational year in scenario S2.1, the RUL of the main frame decreases by 1.1 years and the tower base by 0.9 years.

Moreover, we observe that the RUL for the blade root and the rotor shaft increases in the two scenarios S2.4 and S2.5.

## 3.2 Annual energy production

The annual energy production (AEP) of the wind turbine is calculated using the power curve specified by the turbine manufacturer and the wind speed distribution. The effect of imbalance on the AEP is assessed by comparing scenario D with the imbalance scenarios (Fig. 7). All imbalance scenarios lead to a decrease in AEP compared with the design scenario D. The largest losses amount to 1.2% and occur in scenario S2.1.

## 4 Discussion

### 4.1 Effect of pitch misalignment on components

The previous section shows that different component groups are affected to a different degree by the aerodynamic imbalance. This observation is explained by the type of loading to which the different components are subjected. The components rotating through the gravity field, i.e., in the vicinity of the blade root connection and the rotor shaft, are mainly designed to withstand

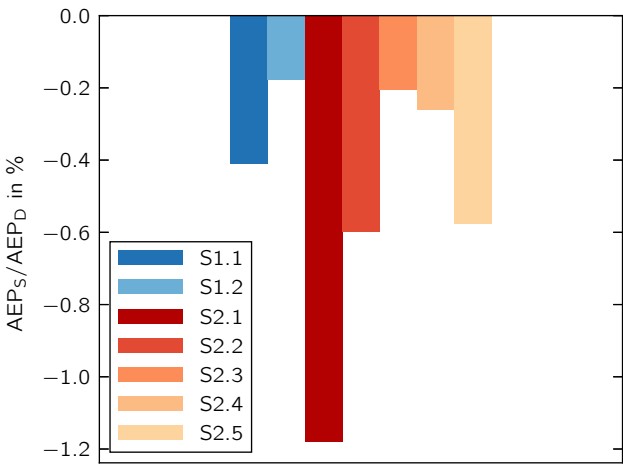

**Figure 7.** Annual energy production in imbalance scenarios (AEP$_S$) compared to design situation (AEP$_D$).

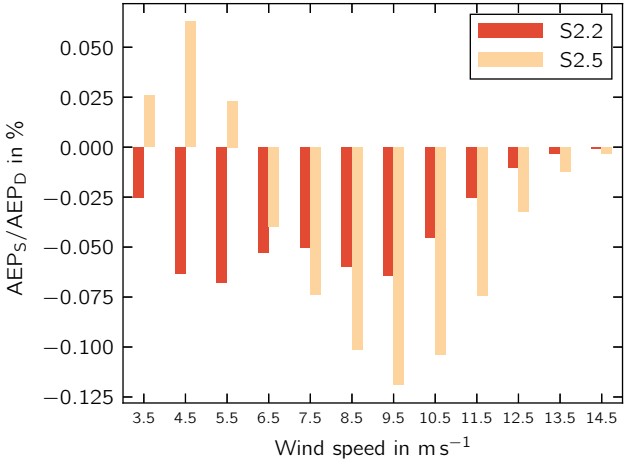

**Figure 8.** Annual energy production per wind speed bin in imbalance scenarios (AEP$_S$) compared to design situation (AEP$_D$).

the alternating inertial loads. The edge-wise blade root bending moment, for example, is to a large extent dominated by the blade mass. Aerodynamic loads and thus an aerodynamic imbalance do therefore not significantly affect the RUL of the components rotating through the gravity field. In addition, the aerodynamic imbalance results from the loading of all three blades and has therefore no significant effect on the root connection of a single blade.

Both aerodynamic loads and the aerodynamic imbalance, in particular, have a larger effect on the rotor shaft and the components in the hub-to-shaft connection than on the blade root. For the non-rotating components, i.e., the main frame, and tower base, the effect of the aerodynamic imbalance is observed clearly. The imbalance excites the turbine in the rotational frequency and its harmonics. Hence, the decrease in RUL is greatest for the non-rotating components.



### 4.2 Effect of pitch misalignment direction

The largest decrease in RUL is observed in scenarios S2.1, S2.2, S2.4, and S2.5 where either one or two blades are misaligned
collectively with an imbalance of $\Delta\theta = 2°$. A misalignment toward stall in scenario S2.1 reduces the RUL more severely than
a misalignment toward feather in scenario S2.5, especially at the tower base. This can be explained by the fact that a pitch
misalignment toward feather leads to a decrease in rotor thrust. While the RUL decreases at the tower base in both scenarios
S2.1 and S2.5 as a result of imbalance, the overall load level tends to be lower in S2.5. The RUL can also increase, however,
in scenarios with an aerodynamic imbalance toward feather, as S2.4 and S2.5. This is because the moment vector amplitude,
which acts on the rotor shaft within the rotor plane, decreases when compared to the design scenario D.

The difference between S2.2 and S2.4 is not as obvious but becomes clear at the tower base, where the effect is amplified by
the lever arm of the tower. The equal distribution scenario S2.3 is overall the least severe scenario as the difference between
aerodynamic loads of the three blades are not as large as in the other scenarios.

The effect that the imbalance severity has on the different component types can also be quantified by comparing the situations
for $\Delta\theta = 1.0°$ with those for $\Delta\theta = 2.0°$, i.e., S1.1 with S2.1 and S1.2 with S2.2. For the rotating components, the losses in
RUL scale with a factor of $1.7$ to $1.9$ with $\Delta\theta$. For the non-rotating components, the losses in RUL scale with a factor of $2.9$ to
$3.2$ with $\Delta\theta$. This observation points to a non-linear relationship between imbalance severity and loss in RUL.

### 4.3 Annual energy production losses

In all cases, the decrease in RUL for the critical components is accompanied by a decrease in AEP. In scenario S2.1, the largest
losses in AEP coincide with the largest losses in RUL.

In the two scenarios S2.2 and S2.5, the AEP losses amount to 0.6%. In S2.5, two of the rotor blades have a misalignment
of $\theta = 2.0°$. In S2.2, only one blade has a misalignment of $\theta = -2.0°$. Hence, a single blade with negative pitch misalignment
has the same adverse effect on AEP as two blades with positive pitch misalignment. This observation can be explained by
evaluating the AEP differences in each wind speed bin (Fig. 8). Because of the minimum rotor speed of the wind turbine, it
is not operated at optimum tip speed ratio at low wind speeds. The misalignment toward stall leads to an increase in power
coefficient at low wind speeds and thus to an increase in AEP at wind speeds between $3.5\,\mathrm{m\,s^{-1}}$ and $6.0\,\mathrm{m\,s^{-1}}$. At wind speeds
higher than $6.0\,\mathrm{m\,s^{-1}}$ and the rated wind speed, the AEP difference becomes negative. For scenario S2.2, the AEP difference
is negative for all wind speeds but the maximum loss is not as large as in S2.5. Ultimately, however, the integrals of the AEP
over the wind speeds of the two scenarios show a similar loss in AEP. Hence, the distribution of wind speeds has an impact on
the effect of blade angle misalignment on the AEP and the results are expected to differ at a different site.

### 4.4 Representativity of simulations

The scenarios simulated represent the derived aerodynamic imbalance of the measured data (Fig. 5) for $1° < \Delta\theta \le 2°$, which
cover 35.3% of the imbalance situations not accepted by the standards and guidelines. 2.6% of the turbines showed an imbal-
ance of $5° < \Delta\theta > 2°$, which is expected to further reduce the RUL.





The scenarios simulated represent an absolute pitch angle misalignment of one blade of $-2° \leq \theta \leq 2°$ (Table 1). It must be noted, however, that the measured pitch angle misalignment of one blade can be $\theta > 4°$ or $\theta < -5°$ (Fig. 4). The absolute pitch angle misalignment is expected to have a negligible effect on aerodynamic imbalance, but can lead to a greater ($\theta < 0°$ toward stall) or smaller ($\theta > 0°$ toward feather) rotor thrust.

### 4.5 Representativity of measurements

The pitch angle misalignment data was determined from the pressure side of the airfoil at the maximum chord cross-section. Considering the maximum chord position has two advantages: (i) the accuracy of the angle calculation is higher and (ii) the effect on the blade deflection and twist due to gravity loads and aerodynamic loads during the measurement is lower when compared to a cross-section further outboard with a shorter chord length.

  It must be noted, however, that most of the aerodynamic loads are generated at the outboard blade portion. Thus, the measured
absolute twist angle relative to the target at this blade portion is of interest for the optimum absolute blade pitch angle alignment. This task would be rather challenging due to there being other causes for the deviation between the actual in-field blade geometry and the target blade design geometry, e.g., manufacture-induced blade distortions (Rosemeier and Saathoff, 2020b).

### 5 Conclusions

The individual blade pitch angle misalignments of 195 wind turbines of different types were measured using a laser-optical
method. From the empirical data, aerodynamic imbalance scenarios, which represent 35.3% of the measured imbalance situations not accepted by the standards and guideline, were derived and assessed by means of aeroelastic simulations.

  The RUL of the turbine served as a metric to quantify the effect of the aerodynamic imbalance. We have shown that the aerodynamic imbalance reduced the RUL in most imbalance scenarios compared to the design situation. Rotating components, i.e., in the vicinity of the blade root and shaft-to-hub connection, were affected less by the imbalance than non-rotating components,
i.e., main frame and tower base, which became limiting for the RUL of the use case turbine in an imbalance situation toward stall. Pitch angle misalignment toward stall had a more severe impact on RUL than misalignment toward feather. Depending on the turbine component, we can observe different non-linear relationships between imbalance severity and loss in RUL.

  The AEP can increase or decrease depending on the wind speed and the direction of misalignment. The total energy production across all wind speeds was always negative, however. The scenario leading to the highest loss in RUL also led to the
highest loss in AEP.

*Data availability.* The data presented in the figures is available at https://doi.org/10.5281/zenodo.4748549 (Saathoff et al., 2021).



*Author contributions.*  BR and MS modeled the use case turbine and its controller, conducted the aeroelastic load simulations, and the fatigue analysis of the turbine components. MR initiated this research and wrote the paper together with MS. TK analyzed the measurement data. The four authors assessed the lifetime extension of the turbine together.

*Competing interests.*  The authors declare that they have no conflict of interest.

*Acknowledgements.*  We acknowledge the support of P. E. Concepts GmbH. Moreover, we would like to thank WIND-consult GmbH for sharing their measurement data for this research.





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
