# Peer review of "Effect of individual blade pitch angle misalignment on the remaining useful life of wind turbines"

_Wind Energy Science, 2021_

## Referee Comment (RC2)

[referee-annotated manuscript omitted]

---

## Author Response (AR2)

**AUTHORS' RESPONSE**

M. Saathoff, M. Rosemeier, T. Kleinselbeck and B. Rathmann

August 16, 2021

Dear Editor,

The manuscript wes-2021-42 entitled "Effect of individual blade pitch angle misalignment on the remaining useful life of wind turbines" submitted to WES Journal, has been revised. This document summarizes the actions we have taken with respect to the referees' comments received until 2021-07-20.

We appreciate the referees' comments very much and are grateful for the helpful suggestions.

This document lists our actions and then repeats the referees' comments referring to the actions taken. References to comments refer pages and lines in the submitted revision 4 (wes-2021-42.pdf).

If not explicitly stated in the actions, the references to lines are valid with the attached document showing the differences between revision 4 submitted and revision 6 (diff_main_rev06_vs_rev04.pdf).

With kind regards,

The Authors

**1  ACTIONS**

A.01 (l. 61) Reworded sentence to make it applicable to the filtered data set.

A.02 (l. 61f) Added explanation how the accuracy was derived.

A.03 (l. 65f) Removed sentence which was not applicable to the filtered data set.

**2  COMMENTS**

**2.1  REVIEWER 1**

The paper under review presents results of a measurement campaign on the prevalence of pitch misalignment in operating win turbines and of a numerical case study to assess effects of pitch misalignment on the RUL of structural components. To the reviewer's knowledge this study is unique and provides novel and relevant knowledge on prevalence and effects of pitch misalignement.

The overall quality of research is high. The state-of-the-art is reviewed thoroughly and relevant references on academic studies and industrial practices are included. The methodology of the numerical case study is in accordance with best practices in aeroelastic simulation and RUL calculation and is presented concisely. The author's conclusions are supported sufficiently by the presented data.

The paper is accepted as is.

**2.2  REVIEWER 2**

The authors present the effect of pitch misalignment on lifetime of components and annual energy production through aero-elastic simulations of an example wind turbine. They have used a data set from laser-optic pitch angle misalignment measurements to determine groups of level of misalignment and imbalance scenarios. The main content of the paper, however, are aero-elastic simulations of a S70 turbine with OpenFAST using the defined imbalance scenarios as input. The simulation results were post-processed to obtain remaining useful lifetime (RUL) of blade root, blade bolts, hub. rotor shaft, main frame, and tower base for the different imbalance scenarios, as well as annual energy production losses.

The paper is clearly written. It contains an appropriate literature review, results are presented and discussed well. The topic is of relevance for the wind industry as aerodynamic imbalance often occurs. The authors claim that their scientific novelty is the effect of pitch misalignment on RUL. Although the effect on RUL may not have been shown in scientific literature, the effect on loads has been discussed in several studies (as mentioned by the authors). Also the applied methods (aero-elastic simulations, RUL calculations) do not contain any novelty. The database of measured pitch misalignments is valuable, but the methodology for generating the database is neither validated nor does it play a significant role in the paper (only used for definition of imbalance scenarios).

At current stage, the content of this paper does not contain sufficient new scientific ideas, analyses, or data for a publication in WES. I recommend a major revision of the paper to improve the scientific relevance of the paper. Please also see further comments in the pdf, listed below.

RC2.01 (l.45ff): "To the authors' knowledge, the effect of the aerodynamic imbalance on the RUL has been mentioned in literature, but not been quantified. To this end, this research analyzes empirical data of pitch angle misalignments encountered in the field. This analysis is then used to derive representative aerodynamic imbalance scenarios. As a use case, a commercial turbine sited in Northern Germany is chosen for the assessment of the RUL of the turbine. "

Scientific novelty is limited. Can you improve this? Interesting to see would be, for instance, validation of the simulation results with measurement data or further studies on generalization of results (sensitivity to case study, turbine type, etc.).

→ *The scientific novelty of this research is basically the quantification, i.e., the substantiation by numbers, of the relative difference between the component's loss in remaining useful life (RUL) as a consequence of an aerodynamic imbalance as typically measured in field. We think our contribution comprises a substantial methodology and analysis which illustrates the relevance of this problem for the application in industry.*

*The comparison of the simulation results with measurement data, e.g., the measurement of wind and strain histories, would be of added value to validate the generic model used in this study. However, using generic turbine models, which are typically not necessarily validated by measurements, in the context of a lifetime extension assessment is generally accepted in industry and upcoming standards (draft of IEC 61400-28). However, the outcome of this study of a validated turbine model is expected to be similar, since mainly the relative difference between the component's loss in remaining useful life is of interest. To conclude, we can assume that the generic model used is representative to answer the research question.*

*We agree that the results are difficult to generalize because each turbine type may react differently at a specific site or at a specific location within a farm. In fact, this study needs to be considered as an example use case to illustrate possible consequences to industry applications. We would like to point out again that the chosen use case represents well the average of the pitch-controlled turbines (power class) and sites of the empirical data set.*

*At least, the sensitivity of the generic turbine model toward the aerodynamic imbalance $\Delta\theta$, i.e., $\Delta\theta = 1°$ and $\Delta\theta = 2°$, was investigated in this study.*

RC2.02 (l.61f): "the angle between the three blades within the rotor could be determined with an accuracy of $\pm 0.1°$."

How is this accuracy determined? What is the accuracy of the calculation of the individual pitch misalignment?

→ *See A.02.*

RC2.03 (l.63): "the target pitch angle of the turbine's blade type were known"

How often was this known? How often was the assumption 'mean across blades' taken?.

$\rightarrow$ *For the filtered data set shown in this study the target pitch angle was always known. See A.01 and A.03.*

RC2.04 (l.114ff) "The tower structure was modeled taking into account parameters from the type certificate and related documents of the S70 turbine (ABH, 1999). The structural and aerodynamic properties of the SSP34 rotor blade type were considered in the model. A generic proportional-integral (PI) controller was applied for the pitch and generator torque controls. The maximum pitch speed of the S70 turbine was chosen to be $5\,^{\circ}\,\mathrm{s}^{-1}$. The electrical pitch drives were modeled with a second order lag element with a corner frequency of $1.2\,\mathrm{Hz}$ and a damping ratio of $\delta = 0.8$."

How sensitive are the results to the turbine model?

$\rightarrow$ *We agree that results of a sensitivity study on model parameters would be an interesting extent to the content of the paper. We expect, however, that a sensitivity study would not further substantiate the main results of this paper, i.e., the relative difference between the component's loss in RUL as a consequence of an aerodynamic imbalance. This is justified by the state-of-the art methodology of a relative load comparison applied to assess the RUL, see Eq. 1. The nature of this method is that model uncertainties are canceled out because the same model is used for the site simulation (numerator in Eq. 1) as well as for the design simulation (denominator in Eq. 1).*